# Discovery of diarylpyrimidine derivatives bearing piperazine sulfonyl as potent HIV-1 nonnucleoside reverse transcriptase inhibitors

Xiangyi Jiang [1], Boshi Huang[1], Shawn Rumrill[2,3], David Pople [2,3], Waleed A. Zalloum [4], Dongwei Kang [1,5], Fabao Zhao[1], Xiangkai Ji[1], Zhen Gao[1], Lide Hu[1], Zhao Wang[1], Minghui Xie[1], Erik De Clercq [6], Francesc X. Ruiz[2,3✉], Eddy Arnold[2,3✉], Christophe Pannecouque[6✉], Xinyong Liu[1,5✉] & Peng Zhan [1,5✉]

HIV-1 reverse transcriptase is one of the most attractive targets for the treatment of AIDS. However, the rapid emergence of drug-resistant strains and unsatisfactory drug-like properties seriously limit the clinical application of HIV-1 non-nucleoside reverse transcriptase inhibitors (NNRTIs). Here we show that a series of piperazine sulfonyl-bearing diarylpyrimidine-based NNRTIs were designed to improve the potency against wild-type and NNRTI-resistant strains by enhancing backbone-binding interactions. Among them, compound **18b1** demonstrates single-digit nanomolar potency against the wild-type and five mutant HIV-1 strains, which is significantly better than the approved drug etravirine. The co-crystal structure analysis and molecular dynamics simulation studies were conducted to explain the broad-spectrum inhibitory activity of **18b1** against reverse transcriptase variants. Besides, compound **18b1** demonstrates improved water solubility, cytochrome P450 liability, and other pharmacokinetic properties compared to the currently approved diarylpyrimidine (DAPY) NNRTIs. Therefore, we consider compound **18b1** a potential lead compound worthy of further study.

[1] Department of Medicinal Chemistry, Key Laboratory of Chemical Biology (Ministry of Education), School of Pharmaceutical Sciences, Cheeloo College of Medicine, Shandong University, 44 West Culture Road, Jinan 250012 Shandong, PR China. [2] Center for Advanced Biotechnology and Medicine, Rutgers University, Piscataway, NJ 08854, USA. [3] Department of Chemistry and Chemical Biology, Rutgers University, Piscataway, NJ 08854, USA. [4] Department of Pharmacy, Faculty of Health Science, American University of Madaba, P.O Box 2882 Amman 11821, Jordan. [5] China-Belgium Collaborative Research Center for Innovative Antiviral Drugs of Shandong Province, 44 West Culture Road, Jinan 250012 Shandong, PR China. [6] Rega Institute for Medical Research, Laboratory of Virology and Chemotherapy, K.U.Leuven, Herestraat 49 Postbus 1043 (09.A097), B-3000 Leuven, Belgium. ✉email: fr193@cabm.rutgers.edu; arnold@cabm.rutgers.edu; christophe.pannecouque@rega.kuleuven.be; xinyongl@sdu.edu.cn; zhanpeng1982@sdu.edu.cn

Acquired immunodeficiency syndrome (AIDS) is caused by the human immunodeficiency virus (HIV)[1]. Nearly 38.4 million people worldwide were infected by HIV in 2021, which remains a pandemic health issue [https://www.unaids.org/en (EB/OL)]. HIV-1 reverse transcriptase (RT) has important biochemical functions for viral replication as RNA/DNA-dependent DNA polymerase and ribonuclease H (RNase H). Thus, it has been considered as one of the most attractive targets for the treatment of AIDS[2]. Currently, RT inhibitors are mainly divided into nucleoside RT inhibitors (NRTIs) and non-nucleoside RT inhibitors (NNRTIs)[3]. As allosteric inhibitors, NNRTIs interfere with the normal function of RT by binding to the NNRTI binding pocket (NNIBP) that is located about 10 Å from the polymerase active site[4]. NNRTIs have been key components in highly active antiretroviral therapy (HAART) due to their promising anti-HIV-1 activities, high specificity, and relatively low toxicity[5,6].

As shown in Fig. 1, Nevirapine (NVP, 1), efavirenz (EFV, 2), etravirine (ETR, 3), rilpivirine (RPV, 4) and doravirine (DOR, 5) are NNRTIs that have been approved by the U.S. FDA for the treatment of AIDS[7]. NNRTIs have a relatively low genetic barrier because of their allosteric binding, which leads to rapid emergence of drug-resistant strains during their clinical applications[8]. For example, the first-generation NNRTIs NVP and EFV, have shown dramatically reduced activities against the mutant strains K103N, Y181C and L100I[9,10]. However, the second generation of NNRTIs, including ETR, RPV and DOR, demonstrated promising activities against early NNRTI-resistant mutations. Nevertheless, some new resistant strains have been selected by the second generation of NNRTIs, such as K103N/Y181C and F227L/V106A for ETR, E138K and F227C for RPV, and V106A and F227L for DOR[11,12]. ETR and RPV (diarylpyrimidine, DAPY) suffer from poor solubility (ETR, ≪1 µg/mL at pH 7.0; RPV, 20 ng/mL at pH 7.0), which affects their pharmacokinetic (PK) properties[13]. Nevertheless, RPV and other DAPY compounds such as dapivirine (R147681) compensate for low solubility through formation of hydrophobic aggregates that improve their bioavailability[14], a phenomenon also observed in other relevant cases[15,16]. In addition, ETR and RPV are inhibitors of cytochrome P450 (CYP) enzymes, which can decrease their effective dose, sometimes necessitating co-formulation with PK enhancers such as ritonavir to compensate[17]. Therefore, there is a pressing need to develop novel NNRTIs with improved anti-HIV-1 activities against resistant mutant strains and improved drug-like properties.

The cocrystal structure of ETR/RT (PDB code: 3MEC, Fig. 2a) indicated that the aminobenzonitrile moiety (right-wing) acted on a rather plastic "groove", namely tolerant region I of the HIV-1 NNIBP[18,19]. The tolerant region I, which is formed from V106, F227, L234, P236 and Y318, is a modifiable chemical space that can accommodate various substituents[20]. Taking ETR as the lead compound, our previous efforts have identified a variety of DAPY-typed NNRTIs by exploiting the tolerant region I[19–23]. Particularly, compound BH-11c (6) revealed remarkable anti-HIV-1 activities

against the wild-type (WT) and several single mutant strains, including K103N, E138K and Y181C[21]. In addition, BH-11c demonstrated significantly enhanced water solubility (33.4 µg/mL at pH 7.0) and safety profiles compared to ETR. However, BH-11c did not show satisfactory activities toward F227L/V106A and K103N/Y181C double mutant HIV-1 strains. Hence, it is worth conducting systematical optimization based on BH-11c to yield novel NNRTIs with improved drug-resistance profiles.

Particularly, the F227L/V106A mutation has gained attention due to its resistance to the current NNRTIs, as well as BH-11c[11,21]. A known effective medicinal chemistry strategy to overcome drug resistance is to establish interactions between the ligand and main chain atom(s) of the surrounding amino acid residues[24,25]. Significantly, there is an extended channel in the tolerant region I, consisting of F227 and P236, that leads to the solvent[26–28]. In order to better understand the interactions between BH-11c and the NNIBP of HIV-1 RT (PDB code: 3MEC)[29], molecular docking studies were performed using the software Surflex-Dock SYBYL-X 2.0[30], and the results were shown in Fig. 2c by PyMOL (http://www.pymol.org/). In the presence of the methylene group, there is a bond angle of 114.2° between piperazinyl and phenyl rings in the right wing of BH-11c (Fig. 2b). This resulted in the sulfonyl group of BH-11c being 4–6.7 Å away from the main chains of F227, P236 and L234, which was too far to interact. Based on this analysis, we designed a series of HIV-1 inhibitors and the docking results of representative compound 18a1 are shown in Fig. 2c. We speculated that the removal of the methylene group could shorten these distances (~3 Å, Fig. 2c), which potentially form additional interactions with F227, P236 and L234. A nitrogen atom, cyano (CN) or trifluoromethyl group was introduced at position 3 of the right phenyl ring, with the aim of establishing additional interactions with V106. A cyano or cyanoethylene (CV) group was introduced as a privileged fragment in the left-wing. We postulated that such structural optimizations derived from BH-11c could generate additional interactions (especially backbone-binding interactions) with the residues in tolerant region I.

This research focuses on the design, synthesis, anti-HIV evaluation and preliminary structure-activity relationships (SARs) of DAPY-typed derivatives. Furthermore, the co-crystal structure determination and molecular dynamics (MD) simulation were utilized to investigate interaction modes between a representative compound and the binding pocket. Finally, drug-like properties, including water solubility, cytochrome P450 (CYP) inhibition, and PK properties were investigated in detail.

## Results

**Synthesis and characterization for the target compounds.** As shown in Fig. 3, 1-Boc-piperazine was reacted with 2-bromo-5-nitropyridine, or 2-bromo-5-nitrobenzonitrile, or 1-bromo-4-nitro-2-(trifluoromethyl)benzene at 120 °C to obtain 8a–c, which

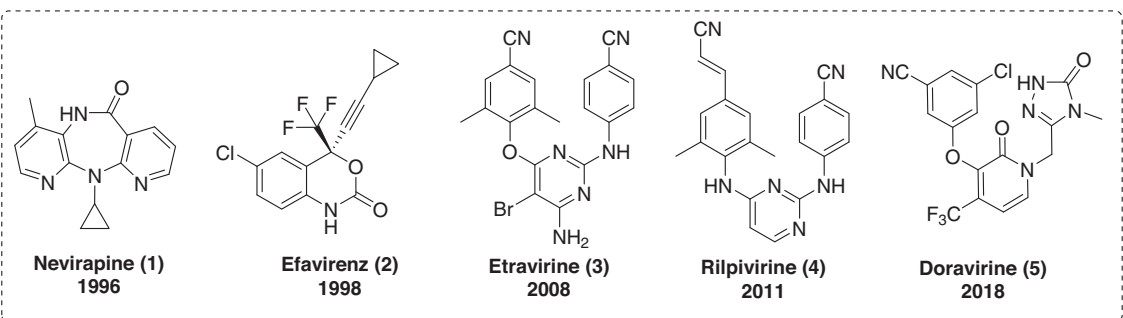

**Fig. 1 Chemical structures of NNRTI drugs approved by U.S. FDA.** Nevirapine (NVP, 1), efavirenz (EFV, 2), etravirine (ETR, 3), rilpivirine (RPV, 4) and doravirine (DOR, 5).

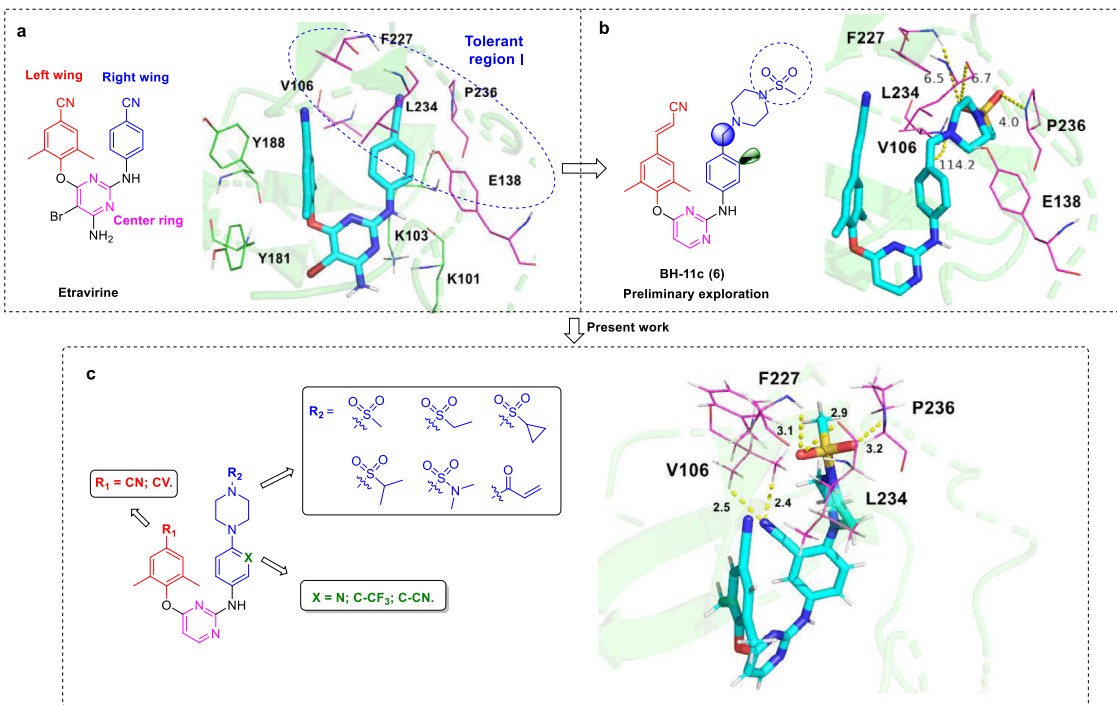

**Fig. 2 Design and optimization of DAPY-typed HIV-1 NNRTIs targeting the main chains of the residues in tolerant region I. a** The cocrystal structure of ETR/RT (PDB code: 3MEC). **b** Predicted binding mode of **BH-11c** bound to the NNIBP (PDB code: 3MEC). **c** Predicted binding mode of newly designed representative compound **18a1** bound to the NNIBP (PDB code: 3MEC). The docking was performed using the Surflex-Dock of SYBYL-X 2.0, and the results were shown by PyMOL (http://www.pymol.org/). The distance between ligands and residues is indicated by dashed line. The bond angle is indicated by an arc dashed line.

**Fig. 3 The synthetic route of 9a–c.** (i) 1-Boc-piperazine, $K_2CO_3$, DMF, 120 °C; (ii) $H_2$, Pd/C, r.t.

were reduced under hydrogen atmosphere to yield intermediates **9a–c**[31,32]. In Fig. 4, the intermediate **11** or **12** was obtained by the reaction of 2,4-dichloropyrimidine (**10**) with 4-hydroxy-3,5-dimethylbenzonitrile or (*E*)-3-(4-hydroxy-3,5-dimethylphenyl) acrylonitrile[33]. Next, **11** or **12** was treated with well-prepared compounds **9a–c** to provide the corresponding key intermediates **13a–c** or **14a–c** *via* a Buchwald-Hartwig coupling reaction[23]. Subsequently, the Boc group of compounds **13a–c** or **14a–c** was removed in the presence of trifluoroacetic acid to offer compounds **15a–c** or **16a–c**[34]. Finally, the target compounds **17a(1–5)**, **17b(1–5)**, **17c(1–5)**, **18a(1–6)**, **18b(1–6)**, or **18c(1–6)** were synthesized by reacting intermediates **15a–c** or **16a–c** with different sulfonyl chlorides or acryloyl chloride, respectively[35].

**Anti-HIV activities and SAR analysis.** The newly synthesized DAPY-typed derivatives were evaluated for their cytotoxicity and activities against HIV-1 WT (IIIB) and K103N/Y181C double mutant (RES056) strains, as well as HIV-2 (ROD) *via* the MTT method in MT-4 cells[36]. The lead compound **BH-11c** and the six

approved drugs zidovudine (AZT), NVP, EFV, ETR, RVP and DOR were used as control drugs. The biological results were determined as $EC_{50}$ (anti-HIV potency), $CC_{50}$ (cytotoxicity) and SI (selectivity index, the ratio of $CC_{50}/EC_{50}$).

As shown in Table 1, the newly synthesized compounds displayed moderate to excellent antiretroviral activity against WT HIV-1 strain with $EC_{50}$ values ranging from 0.65 μM to 0.0014 μM. Among them, compounds **18a1** ($EC_{50}$ = 0.0018 μM) and **18b1** ($EC_{50}$ = 0.0014 μM) were the most potent inhibitors, which are evidently better than six control compounds, i.e., **BH-11c** ($EC_{50}$ = 0.0027 μM), ETR ($EC_{50}$ = 0.0033 μM), EFV ($EC_{50}$ = 0.0037 μM), AZT ($EC_{50}$ = 0.032 μM), NVP ($EC_{50}$ = 0.19 μM) and DOR ($EC_{50}$ = 0.013 μM). In particular, **18b1** displayed antiviral activity similar to the most potent reference RPV ($EC_{50}$ = 0.0010 μM, $CC_{50}$ = 3.98 μM), with relatively low cytotoxicity ($CC_{50}$ = 10.15 μM). In addition, seven compounds **18a(2–5)** and **18b(2–4)** showed extremely potent anti-HIV-1 activities with $EC_{50}$ values ranging from 0.0049 μM to 0.0026 μM, which were equipotent to **BH-11c**, ETR and EFV, and more active than AZT, NVP and DOR. For RES056, two compounds **18b1** ($EC_{50}$ = 0.093 μM) and **18b2** ($EC_{50}$ = 0.089 μM) exhibited the best antiviral activities, which

**Fig. 4 The synthetic route of 17a(1–5), 17b(1–5), 17c(1–6), 18a(1–5), 18b(1–6) and 18c(1–6).** (i) 4-hydroxy-3,5-dimethylbenzonitrile or (*E*)-3-(4-hydroxy-3,5-dimethylphenyl)acrylonitrile, $K_2CO_3$, DMF, 50 °C; (ii) **9a–c**, Pd(OAc)$_2$, xantphos, $Cs_2CO_3$, $N_2$, 1,4-dioxane, 90 °C; (iii) TFA, DCM, r.t.; (iv) different sulfonyl chlorides or acyl chloride, TEA, DCM, 0 °C.

were markedly more potent than **BH-11c** ($EC_{50} = 0.59\,\mu M$), EFV ($EC_{50} = 0.30\,\mu M$) and NVP ($EC_{50} > 15.02\,\mu M$), and were equipotent to ETR ($EC_{50} = 0.076\,\mu M$). As expected, no compounds revealed anti-HIV-2 activities since DAPY-typed derivatives interact with the NNIBP of HIV-1 RT specifically (data were not shown).

According to the results in Table 1, the preliminary SARs and structure-cytotoxicity relationships (SCRs) can be depicted as follow:

1. By comparing series **17** and **18**, it is obvious that the cyanovinyl group at $R_1$ position is more preferred than the cyano group for anti-HIV-1 activity. However, cyanovinyl-containing compounds exhibited enhanced cytotoxicity possibly due to the "Michael addition effect".
2. Next, we turned our attention to substituents at position 3 of the right phenyl ring (X in scheme 2). By comparing the corresponding derivatives in **a**, **b** and **c** series, the potency of compounds against WT HIV-1 was in the following order: **b** series (X = C-CN) ≥ **a** series (X = N) ≥ **c** series (X = C-CF₃). The cytotoxicity of corresponding compounds in **a**, **b** and **c** series was as follows: when $R_1$ is cyano group, **17a** series (X = N) ≥ **17c** series (X = C-CF₃) ≥ **17b** series (X = C-CN), except compounds **17a2/17c2** and **17b6/17c6**; when $R_1$ is cyanovinyl group, **18a** series (X = N) ≥ **18b** series (X = C-CN) ≥ **18c** series (X = C-CF₃).
3. In the case of the terminal substituents on piperazinyl ($R_2$), the order of potency in **17c** series was as follow: **17c1** ($R_2$ = methyl sulfonyl) ≥ **17c2** ($R_2$ = ethyl sulfonyl) ≥ **17c5** ($R_2$ = cyclopropane sulfonyl) ≥ **17c3** ($R_2$ = 2-propane sulfonyl) ≥ **17c4** ($R_2$ = N,N-dimethyl sulfonyl) ≥ **17c6** ($R_2$ = acryloyl). The terminal substituents in other series have a similar effect on the antiviral activity. Moreover, it is apparent that compounds with an acryloyl substituent indicated higher cytotoxicity than other sulfonyl substituents, except compound **17c6**.

According to the anti-HIV-1(WT) results, the promising compounds **18a(1–5), 18b(1–5)** and **18c1** were chosen to further evaluate their inhibitory potency against the prevalent HIV-1 mutations, including L100I, K103N, Y181C, Y188L, E138K, and F227L/V106A. As shown in Table 2, compounds **18b(1–5)** revealed excellent inhibitory effect on these HIV-1 strains. Among them, compound **18b1** which exhibited the most potent antiviral activity

against these HIV-1 variants, apart from Y188L. The results could be summarized as follow:

1. The one of most drug resistant variants F227L/V106A: compounds **18b1** ($EC_{50} = 0.0099\,\mu M$) and **18b2** ($EC_{50} = 0.011\,\mu M$) demonstrated the best antiviral potency, which were ~ 20-fold more potent than lead compound **BH-11c** ($EC_{50} = 0.21\,\mu M$), and better than the reference drugs (ETR, $EC_{50} = 0.025\,\mu M$; EFV, $EC_{50} = 0.26\,\mu M$; NVP, $EC_{50} > 15.02\,\mu M$; RPV, $EC_{50} = 0.082\,\mu M$). In addition, compounds **18b(3–5)** and **18c1** displayed two-digit nanomolar antiviral efficacy equivalent to that of ETR.
2. The L100I mutant strain: compounds **18b(1–3)** and **18b5** demonstrated single-digit nanomolar inhibitory effect, which were more potent than all control compounds except RPV ($EC_{50} = 0.0015\,\mu M$). Furthermore, compounds **18b4** ($EC_{50} = 0.020\,\mu M$) and **18c1** ($EC_{50} = 0.015\,\mu M$) presented the similar potency to the lead compound **BH-11c** ($EC_{50} = 0.014\,\mu M$) and ETR ($EC_{50} = 0.012\,\mu M$).
3. The K103N strain: **18b1** ($EC_{50} = 0.0017\,\mu M$) was substantially more active than the lead compound **BH-11c** ($EC_{50} = 0.0037\,\mu M$) and ETR ($EC_{50} = 0.0043\,\mu M$). Compounds **18b(2–5)** also showed high inhibitory activities with the $EC_{50}$ values ranging from $0.0062\,\mu M$ to $0.0035\,\mu M$.
4. The Y181C strain: compounds **18b1** ($EC_{50} = 0.0064\,\mu M$) and **18b2** ($EC_{50} = 0.0063\,\mu M$) showed 2.8-fold potency enhancement over ETR ($EC_{50} = 0.018\,\mu M$). The antiretroviral activities of **18b3** ($EC_{50} = 0.012\,\mu M$) and **18b4** ($EC_{50} = 0.016\,\mu M$) were superior to the lead compound **BH-11c** ($EC_{50} = 0.027\,\mu M$).
5. The Y188L strain: compounds **18b(1–3)** and **18b5** displayed submicromolar antiviral activities, which were more active than the lead compound **BH-11c** ($EC_{50} = 1.32\,\mu M$), though they were still weaker than ETR ($EC_{50} = 0.025\,\mu M$) and RPV ($EC_{50} = 0.079\,\mu M$).
6. The E138K mutant strain: compounds **18b(1–3)** and **18b5** showed better or equivalent antiviral activities compared to ETR ($EC_{50} = 0.018\,\mu M$), with $EC_{50}$ values between $0.0019\,\mu M$ and $0.0064\,\mu M$. Moreover, **18b1** ($EC_{50} = 0.0064\,\mu M$) was slightly better than **BH-11c** ($EC_{50} = 0.0094\,\mu M$), and was equipotent to RPV ($EC_{50} = 0.0058\,\mu M$).

In summary, **18b1** demonstrated superior potency against these HIV-1 mutant strains over the lead compound **BH-11c**.

**Table 1 Antiretroviral activities against HIV-1 IIIB and RES056 strains, cytotoxicity and selectivity index of the target compounds.**

| Compound | $R_1$ | $R_2$ | $EC_{50}$ ($\mu$M)[a] | | $CC_{50}$ ($\mu$M)[b] | $SI$[c] | |
|---|---|---|---|---|---|---|---|
| | | | IIIB | RES056 | | $III_B$ | RES056 |
| 17a1 | CN | $SO_2CH_3$ | 0.025 ± 0.010 | >28.21 | 28.21 ± 4.36 | 1138 | <1 |
| 17a2 | CN | $SO_2CH_2CH_3$ | 0.025 ± 0.015 | >214.13 | 214.13 ± 17.93 | 8443 | <1 |
| 17a3 | CN | $SO_2CH(CH_3)_2$ | 0.037 ± 0.011 | >6.75 | 6.75 ± 1.30 | 182 | <1 |
| 17a4 | CN | $SO_2N(CH_3)_2$ | 0.037 ± 0.0072 | >54.83 | 54.83 ± 32.48 | 1478 | <1 |
| 17a5 | CN | $SO_2CH(CH_2)_2$ | 0.026 ± 0.0098 | >69.16 | 69.16 ± 54.42 | 2612 | <1 |
| 18a1 | CV | $SO_2CH_3$ | 0.0018 ± 0.00047 | >6.91 | 6.91 ± 1.85 | 3680 | <1 |
| 18a2 | CV | $SO_2CH_2CH_3$ | 0.0038 ± 0.0013 | >4.51 | 4.51 ± 0.31 | 1197 | <1 |
| 18a3 | CV | $SO_2CH(CH_3)_2$ | 0.0039 ± 0.0012 | >3.71 | 3.71 ± 0.70 | 959 | <1 |
| 18a4 | CV | $SO_2N(CH_3)_2$ | 0.0043 ± 0.00065 | >4.54 | 4.54 ± 0.33 | 1063 | <1 |
| 18a5 | CV | $SO_2CH(CH_2)_2$ | 0.0035 ± 0.00069 | >59.27 | ND[d] | NA[e] | NA |
| 17b1 | CN | $SO_2CH_3$ | 0.0084 ± 0.0023 | >76.68 | 76.68 ± 21.55 | 9101 | <1 |
| 17b2 | CN | $SO_2CH_2CH_3$ | 0.0066 ± 0.0017 | >241.49 | >241.49 | >36,585 | NA |
| 17b3 | CN | $SO_2CH(CH_3)_2$ | 0.0098 ± 0.0035 | >195.47 | 195.47 ± 11.64 | 19,985 | <1 |
| 17b4 | CN | $SO_2N(CH_3)_2$ | 0.038 ± 0.015 | >199.73 | 199.73 ± 21.69 | 5252 | <1 |
| 17b5 | CN | $SO_2CH(CH_2)_2$ | 0.0066 ± 0.0021 | >201.75 | 201.75 ± 14.16 | 30,529 | <1 |
| 18b1 | CV | $SO_2CH_3$ | 0.0014 ± 0.00019 | 0.093 ± 0.019 | 10.15 ± 1.82 | 7171 | 110 |
| 18b2 | CV | $SO_2CH_2CH_3$ | 0.0026 ± 0.0011 | 0.089 ± 0.017 | 121.36 ± 22.88 | 47,128 | 1366 |
| 18b3 | CV | $SO_2CH(CH_3)_2$ | 0.0049 ± 0.0016 | 0.21 ± 0.011 | 19.54 ± 13.87 | 3993 | 95 |
| 18b4 | CV | $SO_2N(CH_3)_2$ | 0.0048 ± 0.0011 | 0.47 ± 0.066 | 18.01 ± 14.57 | 3751 | 39 |
| 18b5 | CV | $SO_2CH(CH_2)_2$ | 0.0054 ± 0.0014 | 0.12 ± 0.04 | 195.94 ± 1.85 | 36,051 | 1591 |
| 18b6 | CV | $COCH=CH_2$ | 0.0068 ± 0.0014 | 0.86 ± 0.18 | 4.57 ± 0.58 | 672 | 5 |
| 17c1 | CN | $SO_2CH_3$ | 0.015 ± 0.0051 | >88.04 | 88.04 ± 52.95 | 5951 | <1 |
| 17c2 | CN | $SO_2CH_2CH_3$ | 0.017 ± 0.0037 | >33.72 | 33.72 ± 26.42 | 1949 | <1 |
| 17c3 | CN | $SO_2CH(CH_3)_2$ | 0.048 ± 0.011 | >6.35 | 6.35 ± 1.27 | 134 | <1 |
| 17c4 | CN | $SO_2N(CH_3)_2$ | 0.079 ± 0.029 | >77.31 | 77.31 ± 14.17 | 973 | <1 |
| 17c5 | CN | $SO_2CH(CH_2)_2$ | 0.022 ± 0.0097 | >123.95 | 123.95 ± 24.46 | 5730 | <1 |
| 17c6 | CN | $COCH=CH_2$ | 0.65 ± 0.30 | >98.85 | 98.85 ± 76.78 | 153 | <1 |
| 18c1 | CV | $SO_2CH_3$ | 0.0067 ± 0.0011 | 0.64 ± 0.34 | >218.30 | >32,772 | >341 |
| 18c2 | CV | $SO_2CH_2CH_3$ | 0.013 ± 0.0056 | 0.49 ± 0.14 | >213.08 | >16,827 | >432 |
| 18c3 | CV | $SO_2CH(CH_3)_2$ | 0.028 ± 0.0082 | 4.39 ± 5.37 | ND | NA | NA |
| 18c4 | CV | $SO_2N(CH_3)_2$ | 0.025 ± 0.0072 | 5.10 ± 2.20 | 100.50 ± 54.37 | 3982 | 20 |
| 18c5 | CV | $SO_2CH(CH_2)_2$ | 0.021 ± 0.0061 | 0.43 ± 0.047 | 163.48 ± 21.51 | 7663 | 380 |
| 18c6 | CV | $COCH=CH_2$ | 0.021 ± 0.0050 | >3.78 | 3.78 ± 0.91 | 180 | <1 |
| BH-11c | | | 0.0027 ± 0.0018 | 0.59 ± 0.095 | 5.68 ± 0.47 | 2067 | 10 |
| AZT | | | 0.032 ± 0.0096 | 0.037 ± 0.011 | >7.48 | >234 | >200 |
| NVP | | | 0.19 ± 0.076 | >15.02 | >15.02 | >79 | NA |
| ETR | | | 0.0033 ± 0.00038 | 0.076 ± 0.025 | >4.59 | >1395 | >60 |
| EFV | | | 0.0037 ± 0.00067 | 0.30 ± 0.19 | >6.33 | >1690 | >21 |
| RPV[f] | | | 0.0010 ± 0.00027 | 0.011 ± 0.0080 | 3.98 | 3989 | 371 |
| DOR[g] | | | 0.013 ± 0.004 | 0.015 ± 0.006 | >294 | >21,940 | >20136 |

[a]$EC_{50}$: concentration required to achieve 50% protection of MT-4 cell cultures against HIV-1-induced cytopathicity, as determined using the MTT method (mean ± SD, $n \geq 4$).
[b]$CC_{50}$: concentration required to reduce the viability of mock-infected cell cultures (cytotoxicity, CC) by 50%, as determined using the MTT method (mean ± SD, $n \geq 4$).
[c]SI: selectivity index, the ratio of $CC_{50}/EC_{50}$.
[d]ND: not determined.
[e]NA: no applicable.
[f]Used for comparison. The data were obtained from the same laboratory using the same method (Prof. Christophe Pannecouque, Rega Institute for Medical Research, KU Leuven, Belgium)[4].
[g]Used for comparison. The data were obtained from the same laboratory using the same method (Prof. Christophe Pannecouque, Rega Institute for Medical Research, KU Leuven, Belgium)[7].

Furthermore, **18b1** possessed single-digit nanomolar potency against five mutants including L100I, K103N, Y181C, E138K and F227L/V106A. Also, compound **18b1** is about 3-fold more potent than ETR.

**WT HIV-1 RT inhibition assay.** In order to confirm the drug target of the synthesized DAPY derivatives, representative compounds were selected to test their inhibitory activity against WT HIV-1 RT. As shown in Table 3, these selected compounds exhibited inhibitory effect on HIV-1 RT with $IC_{50}$ values between 0.19 to 0.056 $\mu$M, being evidently better than RVP ($IC_{50} = 0.43$ $\mu$M) and ETR ($IC_{50} = 1.35$ $\mu$M). The results indicated that these compounds target HIV-1 RT, and they could be classified as HIV-1 NNRTIs.

**Crystal structure of HIV-1 RT in complex with 18b1.** We have determined the crystal structure of HIV-1 wild-type (WT) RT in complex with **18b1** at 2.5 Å resolution (Supplementary Table S1 and Supplementary Fig. S1) to gain a thorough understanding of the previously explained SAR. In general, the structure is very similar to previously reported RT-RPV[37] and RT-**25a**[3] structures (Fig. 5a, Supplementary Fig. S1). Notably, though, the cyano group of **18b1** displays hydrophobic contacts with the side chains of V106 and F227, and the sulfonyl group forms hydrogen bonds with the main chains of F227 and L234 (Fig. 5b). The comparison with previous structures shows that the positioning of **18b1** in the NNIBP is similar to RPV, except for the piperazine sulfonyl moiety that protrudes into the tolerant region I—as in the RT-**25a** complex (PDB 6C0N)—provoking the uplift of the loop preceding β9 and the one connecting β10-β11 (Fig. 5c). However, the

**Table 2 Antiretroviral activity of selected compounds against HIV-1 mutant strains.**

| Compound | EC$_{50}$ (µM)$^a$ | | | | | |
|---|---|---|---|---|---|---|
| | L100I | K103N | Y181C | Y188L | E138K | F227L/V106A |
| **18a1** | 0.050 ± 0.0061 | 0.023 ± 0.0067 | 0.10 ± 0.018 | >6.91 | 0.028 ± 0.0055 | 0.35 ± 0.069 |
| **18a2** | 0.056 ± 0.019 | 0.040 ± 0.0072 | 0.11 ± 0.029 | >4.51 | 0.036 ± 0.0081 | 0.38 ± 0.12 |
| **18a3** | 0.080 ± 0.021 | 0.043 ± 0.0075 | 0.11 ± 0.0055 | >3.71 | 0.044 ± 0.011 | 0.41 ± 0.087 |
| **18a4** | 0.080 ± 0.011 | 0.036 ± 0.0020 | 0.14 ± 0.027 | >4.53 | 0.040 ± 0.0099 | 0.28 ± 0.10 |
| **18a5** | 0.084 ± 0.034 | 0.029 ± 0.0020 | 0.090 ± 0.011 | >59.27 | 0.033 ± 0.062 | 0.30 ± 0.061 |
| **18b1** | 0.0037 ± 0.00062 | 0.0017 ± 0.00032 | 0.0064 ± 0.00067 | 0.74 ± 0.21 | 0.0064 ± 0.00054 | 0.0099 ± 0.0016 |
| **18b2** | 0.0044 ± 0.0011 | 0.0035 ± 0.0012 | 0.0063 ± 0.00061 | 0.58 ± 0.12 | 0.0082 ± 0.00076 | 0.011 ± 0.0029 |
| **18b3** | 0.0068 ± 0.0014 | 0.0062 ± 0.0019 | 0.012 ± 0.0025 | 0.94 ± 0.22 | 0.019 ± 0.0077 | 0.028 ± 0.017 |
| **18b4** | 0.020 ± 0.0055 | 0.0061 ± 0.0021 | 0.036 ± 0.0087 | >18.01 | 0.040 ± 0.0054 | 0.027 ± 0.012 |
| **18b5** | 0.0062 ± 0.0012 | 0.0055 ± 0.00043 | 0.016 ± 0.0050 | 0.50 ± 0.025 | 0.019 ± 0.0053 | 0.021 ± 0.019 |
| **18c1** | 0.015 ± 0.0068 | 0.0073 ± 0.0018 | 0.028 ± 0.0029 | 5.46 ± 4.50 | 0.032 ± 0.0049 | 0.019 ± 0.0068 |
| **BH-11c** | 0.014 ± 0.0053 | 0.0037 ± 0.0016 | 0.027 ± 0.0025 | 1.32 ± 0.22 | 0.0094 ± 0.0050 | 0.21 ± 0.095 |
| **NVP** | 2.23 ± 0.98 | 6.04 ± 2.66 | 7.40 ± 1.78 | >15.02 | 0.22 ± 0.12 | >15.02 |
| **ETR** | 0.012 ± 0.0051 | 0.0043 ± 0.0010 | 0.018 ± 0.0038 | 0.025 ± 0.0076 | 0.018 ± 0.0072 | 0.025 ± 0.0072 |
| **EFV** | 0.053 ± 0.028 | 0.12 ± 0.024 | 0.0063 ± 0.0020 | 0.25 ± 0.056 | 0.0065 ± 0.0017 | 0.26 ± 0.10 |
| **RPV$^b$** | 0.0015 ± 0.00 | 0.0013 ± 0.00036 | 0.0047 ± 0.00048 | 0.079 ± 0.00077 | 0.0058 ± 0.00011 | 0.082 ± 0.021 |

$^a$EC$_{50}$: concentration required to achieve 50% protection of MT-4 cell cultures against HIV-1-induced cytopathicity, as determined using the MTT method (mean ± SD, $n \geq 4$).
$^b$Used for comparison. The data were obtained from the same laboratory using the same method (Prof. Christophe Pannecouque, Rega Institute for Medical Research, KU Leuven, Belgium)[4].

removal of the linking carbon present in **25a** (alongside the previously mentioned contacts of the cyano group) accounts for a substantial shift in the piperazine sulfonyl moiety positioning in comparison to the piperidine-linked benzenesulfonamide of **25a**. Comparison of the protein-ligand contacts (Supplementary Fig. S2) indicate that the main interactions are similar for **25a** and **18b1**. A thorough analysis (using http://www.ebi.ac.uk/thornton-srv/databases/cgi-bin/pdbsum/GetPage.pl?pdbcode=index.html) reveals that **18b1** has a larger number of non-bonded contacts, especially in terms of backbone interactions (see log files in the Supplementary Information).

**Molecular dynamics (MD) simulation study**. Subsequently, the MD simulation was conducted to further explain the differences in the inhibitory activity of **18b1** and ETR against RT variants. Three x-ray structures of HIV-1 RTs[3], including K103N RT (PDB code: 6C0O), E138K RT (PDB code: 6C0P) and V106A/F227L RT (PDB code: 6DUF) were downloaded from RCSB PDB (rcsb.org)[38].

Figure 6 shows the root mean square deviation (RMSD) of **18b1** and ETR during 300 ns MD simulation with corresponding RT variants. It is obvious that **18b1** has clustered into distinct conformations, which is clear from the pattern and the range of RMSD values. On the contrary, the RMSD plot for ETR shows a stable binding conformation with protein. The most dominant cluster analysis of **18b1** with corresponding RT variants shows that they have similar binding modes (Fig. 7a, c, e). However, the conformation of piperazinyl and the orientation of methyl sulfone are varied. The cyano group of **18b1** is orientated towards the left wing, except K103N variant, which suggest the unique orientation for the cyano group is caused by the mutation of K103 to N103 when binding this variant. The binding modes of representative structure of the most dominant clusters of **18b1** and ETR bound to the three RT variants are similar regarding the orientation of right and left wings of both compounds in all RT variants binding (Fig. 7).

Figure 7 shows the binding poses of **18b1** and ETR to the binding sites of the K103N, E138K and V106A/F227L RT variants in the most abundant cluster of each binding. Supplementary Table S2 shows detailed interactions, hydrogen bonding and hydrophobic interaction, of both compounds to the three RT variants. The general view of Fig. 7 and Supplementary Table S2 shows that **18b1** forms more interactions with the surrounding amino acids comparing ETR.

Hydrogen bond interactions between **18b1** and ETR and corresponding RT variants were investigated by hydrogen bond analysis implemented in *cpptraj* (Supplementary Table S2). Supplementary Table S2 shows that the NH group on the right wing of **18b1** and ETR form hydrogen bonds with the backbone oxygen of K103 (N103 in K103N). Interestingly, in contrast to the interactions with other variants, both inhibitors contact the backbone HN of K101 in E138K variant. This could be explained by the lost hydrogen bond between K101 and K138 in E138K, as presented in the other RT variants according to the hydrogen bond analysis. This made K101 orientates to a position that can interact with the inhibitor more frequently than other variants.

The other type of hydrogen bond takes place between C-H and oxygen or nitrogen atom[39–42]. It is considered an important bonding force in biomolecules, despite it is a weak interaction. Supplementary Table S2 shows that the oxygen atoms of **18b1** sulfone (O1 and O2) are involved in aliphatic hydrogen bonding with residues F227, L234 and P236 at good frequency. In K103N variant there are interactions to P225, F227, and P236 with sulfone oxygen atoms. Also, it establishes conventional hydrogen bonds with N103, which is not present in the binding of ETR. The interactions of sulfone oxygen atoms with surrounding amino acids added extra bonding forces, which were reflected on the

**Table 3 Inhibitory activity against WT HIV-1 RT.**

| Compound | IC$_{50}$ (µM)[a] | Compound | IC$_{50}$ (µM) | Compound | IC$_{50}$ (µM) |
|---|---|---|---|---|---|
| 17a1 | 0.056 ± 0.0029 | 18a1 | 0.057 ± 0.0028 | RVP | 0.43 ± 0.087 |
| 17b1 | 0.067 ± 0.013 | 18b1 | 0.093 ± 0.017 | EFV | 0.011 ± 0.0023 |
| 17c1 | 0.19 ± 0.021 | 18c1 | 0.14 ± 0.014 | ETR | 1.35 ± 0.35 |

[a]IC$_{50}$: concentration required to inhibit biotin deoxyuridine triphosphate incorporation into HIV-1 IIIB RT by 50% (mean ± SD, $n = 2$).

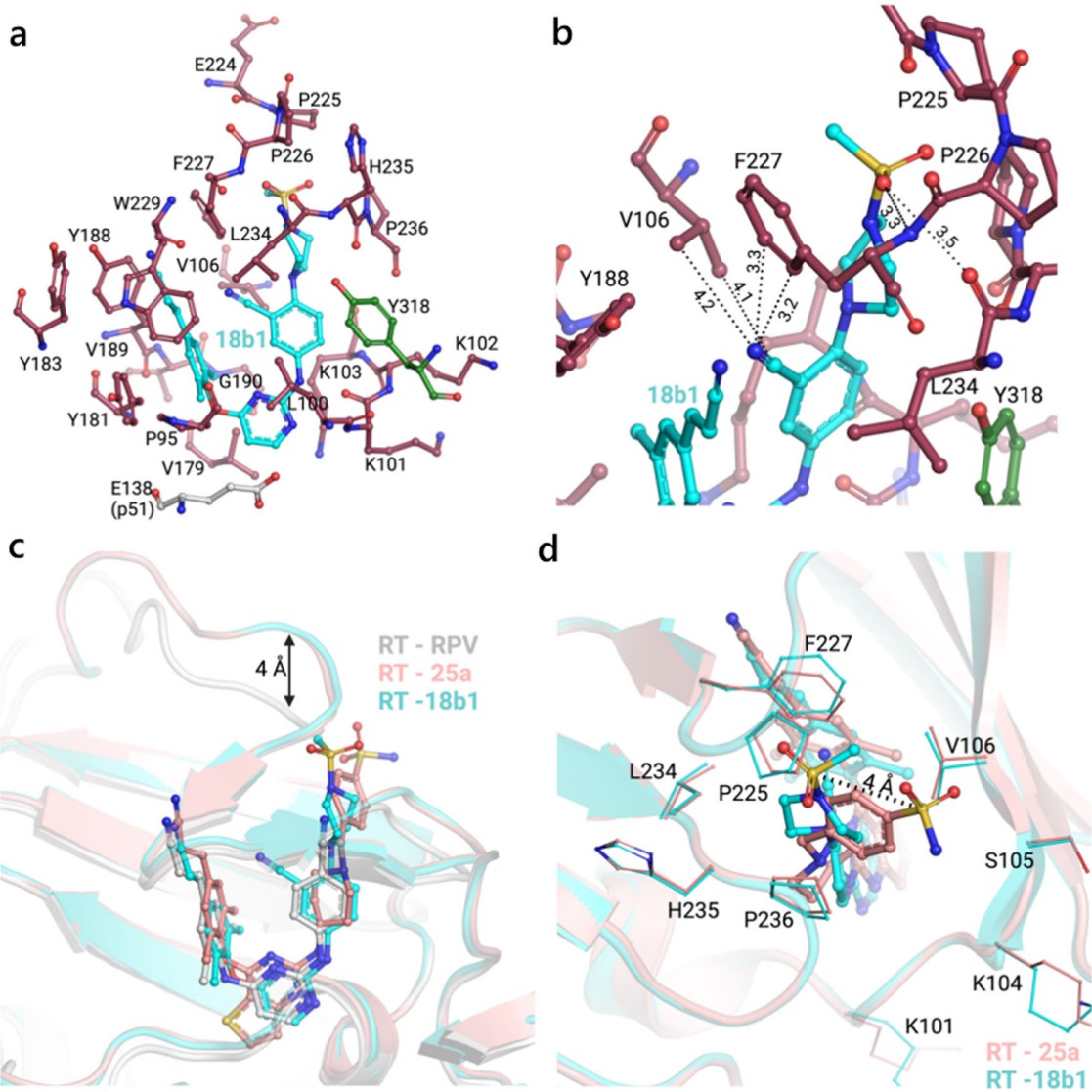

**Fig. 5 Crystal structure of HIV-1 RT in complex with 18b1 (PDB ID: 8FE8). a** 18b1 binding in the NNIBP. Color legend: (i) RT p66 subdomains: palm in dark red, thumb in green; (ii) RT p51 subunit in silver. **b** 18b1 cyano and sulfonyl groups contacts with RT. **c** Comparison of the RT complexes with RPV (PDB 4G1Q) and **25a** (PDB 6C0N). **d** Comparison of the piperazine sulfonyl moiety positioning in the RT complexes with **18b1** and **25a** (PDB 6C0N). Created with PyMOL (http://www.pymol.org/) and with BioRender.com.

activity of **18b1** against each RT variant. Interestingly, the piperazine moiety of **18b1** could forms additional interactions with the backbone atoms (H and O) of L234, H235 and P236 in the three variants frequently (Supplementary Table S2). These interactions are rarely found in the binding of ETR. Forming hydrogen bonds with the backbone atoms has advantage over interactions with the side chain of amino acids since the mutation will have minimal effect on the activity of these inhibitors. This could explain the activity of **18b1** against the mutant variants compared to the activity of ETR.

Hydrophobic interactions were also investigated in Supplementary Table S2. It is clear that the hydrophobic interactions are dominant when binding to RT variants. **18b1** shows interactions with P95 in E138K, whereas it is not present in the interaction of ETR to any RT variant. Interactions with E138 are present between **18b1** and K103N. Also, **18b1** interacts with P236 in all variants. These extra hydrophobic interactions and the above-mentioned hydrogen bonds suggest **18b1** behaves better potency comparing ETR, which is consistent with its experimental activity.

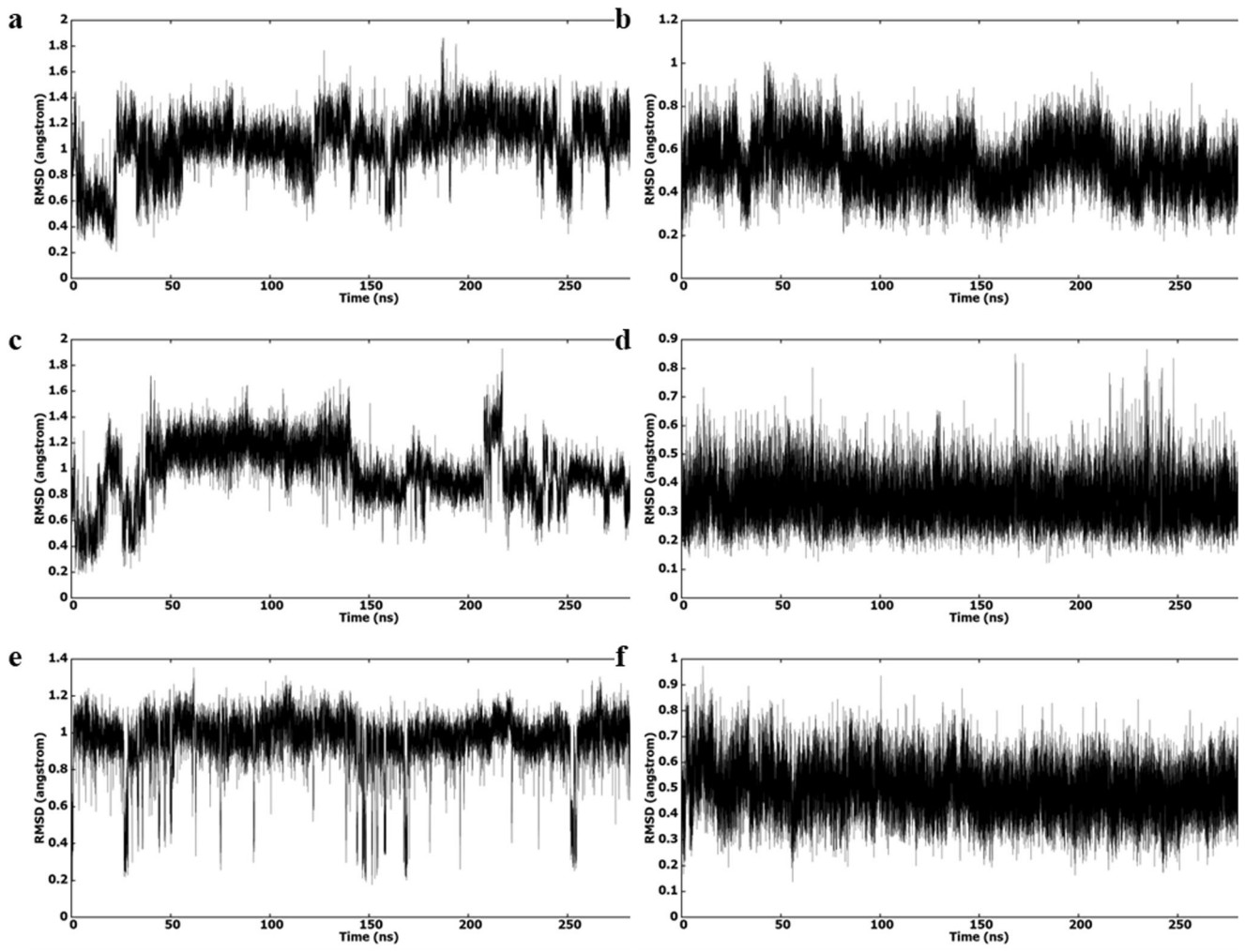

**Fig. 6 RMSD (heavy atoms) in reference to the first frame during the 300 ns MD simulation. a** K103N-**18b1**, **b** K103N-ETR, **c** E138K-**18b1**, **d** E138-ETR, **e** V106A/F227L-**18b1** and **f** V106A/F227L-ETR.

The binding free energies of **18b1** and ETR with corresponding RT variants were calculated using MMPBSA[43] and were represented in Supplementary Table S3. The Van der Waals (vdw) contribution for **18b1** bound to the variants is higher (more negative $E_{vdw}$) than the contribution of ETR, which shows the importance of hydrophobic interactions in their activities. Also, it is clear that the contribution to V106A/F227L is the highest followed by E138K and K103N. The nonpolar contribution to the solvation energy shows favorable binding of both **18b1** and ETR with a clear preference to the binding of **18b1**. The electrostatic contribution was the highest for K103N followed by V106A/F227L and E138K. The electrostatic contribution to the solvation free energy was the highest for V106A/F227L, followed by K103N and E138K, which shows effect on the binding of **18b1** to each variant. Supplementary Fig. S3 shows the free energy decomposition of binding site amino acids, which elucidates amino acid binding energies, individually (vdw, electrostatic, polar solvation and nonpolar solvation energies). The electrostatic contribution of E138 favors the binding of both **18b1** and ETR to all variants except E138K. According to Supplementary Fig. S3, vdw contribution of E138 favors the binding of **18b1** in all variants, except in E138K which slightly favors the binding of ETR. The mutation of E138 to K138 made the contribution of the polar solvation energy favored, where it is negative in E138K and positive in other variants. Inspecting the vdw interactions in Supplementary Fig. S3 shows that the majority of amino acids

favor the binding of **18b1** to all RT variants. Also, the electrostatic energy of most amino acids favors the binding of **18b1** to all RT variants. Furthermore, the nonpolar salvation energy is more negative in the binding of **18b1** to all RT variants.

**Fsp³ value, water solubility, milogP and ligand efficiency.** "Fraction of sp³ carbon atoms" (Fsp³) is used to characterize the carbon saturation of molecules, and it is considered an important parameter for drug-like properties[44,45]. According to the statistics, about 84% of approved drugs possess Fsp³ values above 0.42[19,41]. As indicated in Table 4, ETR suffers from a low Fsp³ value (0.10) due to the presence of multiple aromatic fragments, which led to poor water solubility (<1 μg/mL at pH 7.0) and undesirable milogP (5.03). By introduction of piperazine methylsulfonyl, **18b1** possessed acceptable water solubility (13.46 μg/mL at pH 7.0) and reasonable milogP (4.52) with an improved Fsp³ content (0.27). Furthermore, ligand efficiency (LE) as an emerging index is used to evaluate the balance between biological activity and drug-likeness[46,47]. A promising lead compound is considered to have an LE value above 0.3[48,49], so **18b1** displayed appropriate LE value (0.32) as with ETR (0.41).

**In vitro cytochrome P450 (CYP) enzymatic inhibitory activity.** It is reported that ETR and RPV inhibit CYP enzymes, which could cause potential drug-drug interactions and limit the co-

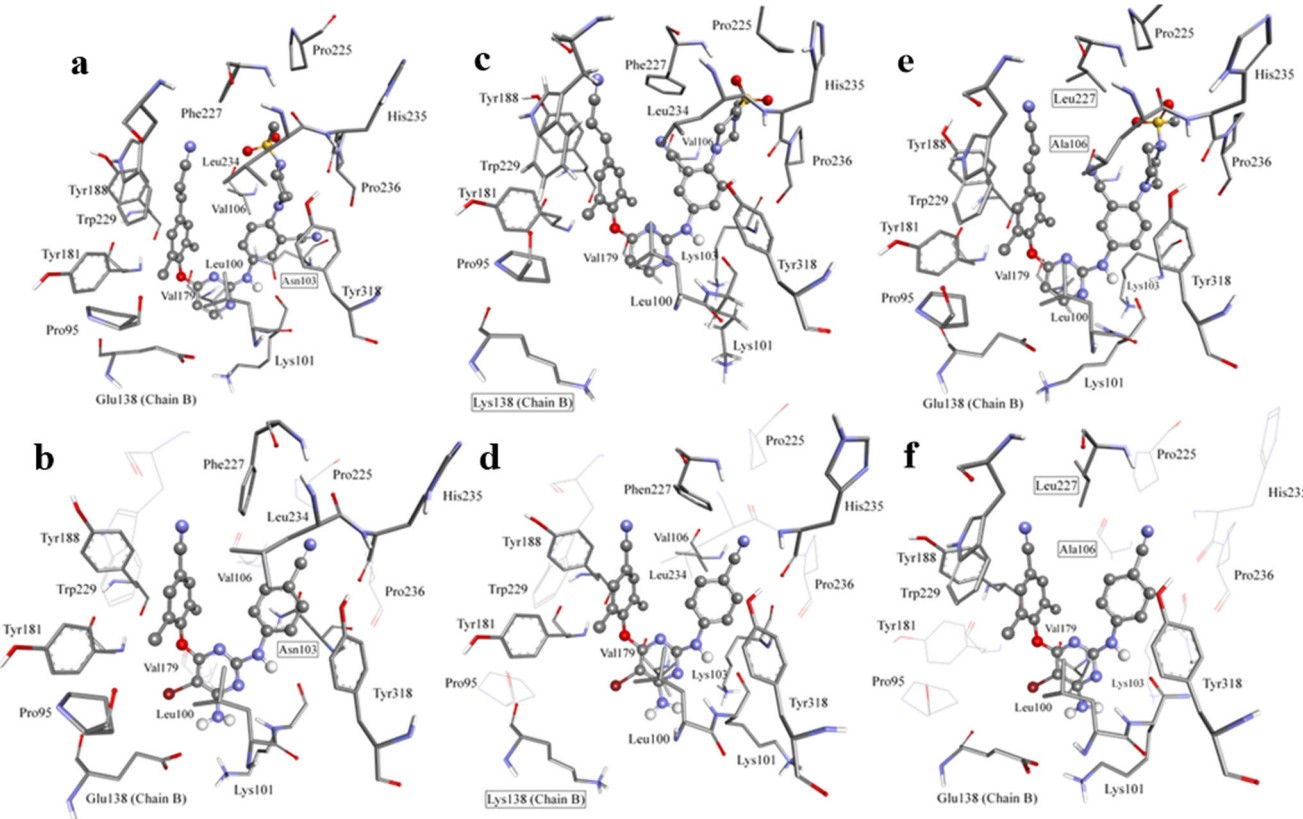

**Fig. 7 Interactions in the representative structures (most abundant cluster) of NNRTIs and HIV-1 RT. a** K103N-**18b1**, **b** K103N-ETR, **c** E138K-**18b1**, **d** E138-ETR, **e** V106A/F227L-**18b1** and **f** V106A/F227L-ETR. Mutated amino acids are presented inside a box. Amino acids that are not involved in interactions with ETR are represented in lines. Inhibitor is represented as balls and sticks.

**Table 4 Fsp$^3$, water solubility, milogP and LE values of 18b1 and ETR.**

| Compound | Fsp$^3$ (>0.42)$^a$ | water solubility (at pH7.0, μg/mL)$^b$ | milogP (<5)$^c$ | LE (>0.3)$^d$ |
|---|---|---|---|---|
| **18b1** | 0.27 | 13.46 ± 0.49 | 4.52 | 0.32 |
| **ETR** | 0.10 | <1$^e$ | 5.03 | 0.41 |

$^a$Fsp$^3$ = the number of sp$^3$ hybridized carbons/total carbon count.
$^b$Measured with HPLC method (mean ± SD, $n$ = 2).
$^c$miLog P = molinspiration predicted Log P.
$^d$LE = calculated by the formula − $\Delta G$/HA (non–hydrogen atom), in which normalizing binding energy $\Delta G = -$ RT ln $K_d$, presuming $K_d \approx EC_{50}$ (III$_B$); $R = 1.987 \times 10^{-3}$ kcal K$^{-1}$ mol$^{-1}$, $T = 298$ K.
$^e$See ref. [21].

administration of multiple drugs[50]. Therefore, **18b1** was assessed for its inhibitory potency against the five main CYP isozymes. ETR and selected inhibitors were used as control. As shown in Table 5, **18b1** exhibited weak inhibitory activity against CYP2C9 (IC$_{50}$ = 5.30 μM) and CYP2C19 (IC$_{50}$ = 8.00 μM), while conversely ETR (IC$_{50}$ = 0.277 μM and 0.496 μM, respectively) and RPV (IC$_{50}$ = 0.346 μM and 0.335 μM, respectively) were sub-micromolar CYP2C inhibitors. On the other hand, **18b1** and ETR demonstrated no obvious inhibitory effect (>10 μM) towards CYP1A2, CYP2D6 and CYP3A4. These results indicate that **18b1** is less likely to cause CYP-mediated drug-drug interactions.

**In vivo pharmacokinetics study.** Compound **18b1** was further selected to evaluate its pharmacokinetics (PK) profile in Sprague-Dawley (SD) rat model. As described in Table 6, **18b1** demonstrated acceptable terminal half-life ($T_{1/2}$ = 2.42 h), moderate clearance (CL = 2.54 L/h/kg) and favorable distribution volume (V = 8.81 L/kg) after an intravenous dose of 2.0 mg/kg. When orally administered at a dose of 10.0 mg/kg, the maximum concentration ($T_{max}$) of **18b1** was 16.05 ng/mL at 3.50 h ($T_{max}$) and the $T_{1/2}$ was 3.05 h. Nevertheless, the oral bioavailability ($F$) of **18b1** was detected to be 1.34%, which requires further optimization, though this is more serious for ETR (undetectable[51]).

## Discussion

In this research, compound **BH-11c** was utilized as a lead from which we designed a series of DAPY-typed NNRTIs, aiming at enhancing backbone-binding interactions with the residues in tolerant region I. Interestingly, the newly synthesized compound **18b1** demonstrated significantly improved antiretroviral activities compared to **BH-11c** against all the tested HIV-1 strains. Furthermore, **18b1** possessed single-digit nanomolar potency against the wild-type and five mutant HIV-1 strains, including L100I, K103N, Y181C, E138K and F227L/V106A. The co-crystal structure indicated that the sulfonyl group of **18b1** formed hydrogen bonds with main-chain atoms of F227 and L234, and the cyano group displayed hydrophobic contacts with side-chain atoms of V106 and F227. Further MD simulation studies were conducted

**Table 5 Effects of 18b1 on inhibition of CYP1A2, CYP2C9, CYP2C19, CYP2D6 and CYP3A4.**

| Compound | IC$_{50}$ (µM) | | | | |
|---|---|---|---|---|---|
| | CYP1A2 | CYP2C9 | CYP2C19 | CYP2D6 | CYP3A4 |
| **18b1** | >100 | 5.30 | 8.00 | 11.3 | 30.7 |
| **ETR** | 7.48 | 0.277 | 0.496 | 12.0 | 41.3 |
| **RPV** | 9.11 | 0.346 | 0.335 | 3.41 | 2.17 |
| *α*-naphthoflavone | 0.322 | | | | |
| Sulfaphenazole | | 0.487 | | | |
| (+)-*N*-3-benzylnirvanol | | | 0.174 | | |
| Quinidine | | | | 0.128 | |
| Ketoconazole | | | | | 0.0427 |

**Table 6 Pharmacokinetics profiles of 18b1$^a$.**

| Parameter | 2.0 mg/kg (i.v.) | 10.0 mg/kg (p.o.) |
|---|---|---|
| $T_{1/2}$ (h) | 2.42 ± 0.34 | 3.05 ± 1.44 |
| $T_{max}$ (h) | 0.08 | 3.50 ± 2.78 |
| $C_{max}$ (ng/mL) | 1233.33 ± 96.09 | 16.05 ± 6.59 |
| AUC$_{0-t}$ (h· ng/mL) | 785.06 ± 73.92 | 45.11 ± 12.92 |
| AUC$_{0-\infty}$ (h· ng/mL) | 792.02 ± 74.88 | 53.21 ± 8.32 |
| V (L/kg) | 8.81 ± 0.72 | |
| CL (L/h/kg) | 2.54 ± 0.25 | |
| F (%) | | 1.34 |

$^a$PK parameter (mean ± SD, $n$ = 3).

to explain the differences in the inhibitory activity of **18b1** and ETR against RT variants. Compared to ETR (<1 µg/mL at pH 7.0), **18b1** displayed improved water solubility (13.46 µg/mL at pH 7.0) with an appropriate LE value (0.32). Moreover, **18b1** revealed significantly lower inhibitory activity than ETR and RPV against CYP2C9 and CYP2C19, indicating that **18b1** was less likely to cause drug-drug interactions. Nevertheless, **18b1** requires further optimization to improve the oral bioavailability (F = 1.34%). Consequently, we consider **18b1** as a promising lead compound worthy of further study.

## Methods

**Molecular simulation methods**. See Section 1 in Supplementary Information.

**Synthetic procedures**. See Section 2 in Supplementary Information. For original spectra of compounds see Supplementary Data 2.

**Anti-HIV activity test studies**. See Section 3 in Supplementary Information.

**HIV-1 RT inhibition assays**. See Section 4 in Supplementary Information.

**HIV-1 RT crystallization and structure determination (including Supplementary Figs. S1–2, Supplementary Table S1)**. See Section 5 in Supplementary Information. For PDB file see Supplementary Data 1.

**Molecular dynamics simulation methods (including Supplementary Fig. S3, Supplementary Tables S2–3)**. See Section 6 in Supplementary Information.

**Water solubility measurements**. See Section 7 in Supplementary Information.

**CYP enzyme inhibition assay**. See Section 8 in Supplementary Information.

**Pharmacokinetics assays**. See Section 9 in Supplementary Information.

**Reporting summary**. Further information on research design is available in the Nature Portfolio Reporting Summary linked to this article.

## Data availability

The authors declare that the data supporting the findings of this study are available within the paper and its Supplementary Information, Supplementary Data 1 and Supplementary Data 2. PDB accession code: crystal structure of HIV-1 RT in complex with the non-nucleoside inhibitor **18b1**, PDB entry: 8FE8.

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

## Acknowledgements

We gratefully acknowledge financial support from the Natural Science Foundation of China (grant numbers 81773794 and 81974507), Science Foundation for Outstanding Young Scholars of Shandong Province (ZR2020JQ31), Foreign cultural and educational experts Project (GXL20200015001), Qilu Young Scholars Program of Shandong University, the Taishan Scholar Program at Shandong Province, NIH grants T32 GM008339 and T32 GM135141 (to S.R.), and NIH grant R01 AI027690 (to E.A.). The authors would like to thank the OpenEye Scientific Software, Inc. for providing a free academic license.

## Author contributions

All authors read and approved the final manuscript. X.L., P.Z. and B.H. are the administrators of the study, they reviewed and edited the manuscript. E.D.C. and C.P. performed the evaluation of compounds activity. W.A.Z. performed molecular docking and molecular dynamic simulation analysis. S.R., D.P., F.X.R. and E.A. prepared recombinant RT protein, solved the RT-18b1 crystal structure and analyzed the data. X.Jiang, X.Ji, L.H., Z.G. performed the design, synthesis and structure confirmation of these compounds. X.Jiang, D.K. and F.Z. designed the pharmacokinetics assays and the CYP450 inhibition assay. Z.W. and M.X. designed the Water solubility measurements.

## Competing interests

The authors declare no competing interests.
