## [Peer Review File · Communications Chemistry]

Reviewers' comments:

Reviewer #1 (Remarks to the Author):

The manuscript entitled "Enhancing Backbone-Binding Interactions: Discovery of Diarylpyrimidine Derivatives bearing Piperazine Sulfonyl as Potent HIV-1 Non-nucleoside Reverse Transcriptase Inhibitors against Wild-Type and F227L/V106A Mutant Viruses" discusses the optimisation of a previous hit compound, BH-11c, ultimately leading to 18b1 through a rather extensive synthetic campaign that was heavily guided by molecular modelling. The final compound, 18b1, demonstrated superior properties to the control compound (ETR), both in terms of potency (especially against problematic resistant strains) and PK properties, which in some categories were superior to that of ETR. In fact, the only major shortcoming of the compound was found to be its poor oral bioavailability (1.34%). Now specifically because of this poor oral bioavailability I feel the authors should change the final line in their abstract which states "Therefore, we consider compound 18b1 a potential drug candidate for the treatment of HIV-1 infections", as with such poor oral bioavailability this is not the case unless the route of administration is planned to be IV (which would be suboptimal). Other than this, the paper is quite well written although it contains many grammatical and technical errors, most of which have been annotated on the attached PDF. In my personal opinion, the modelling section of the paper is too lengthy and in fact detracts from the punchy conclusion of the design and synthesis of 18b1, and the authors may consider shortening this section. In the supplementary section the procedures are adequately described and the compounds adequately characterised.

Therefore I recommend that the article is published after the minor corrections described above and in the attached annotated PDF file.

Reviewer #2 (Remarks to the Author):

Reviewer Summary and Major Claims of the Paper

The authors have designed and synthesized piperazine sulfonyl DAPY analogs as non-nucleoside reverse transcriptase inhibitors (NNRTI). The inhibitors were then evaluated for anti-HIV activity, SAR, inhibition, physicochemical properties such as solubility, mlogP , ligand efficiency, and in vitro CYP inhibition.

Compound 18b1 emerged as a top compound showing activity for several variants of HIV RT. 18b1 also showed improved physicochemical properties (compared to NNRTI etravirine) and overall less inhibition by common CYP isoforms. 18b1 was further analyzed through molecular dynamics (MD) studies to analyze binding conformation and drug-target interactions.

Major comments:

1) The authors use etravirine as a standard NNRTI to compare anti-HIV activity, inhibition, and physicochemical properties with the new piperazine sulfonyl DAPY analogs. Why did the authors not include a comparison with rilpivirine? Rilpivirine is another DAPY included in many 2 combination therapies Cabenuva and Complera. It would be more compelling for the authors to compare values for the piperazine sulfonyl DAPY analogs with rilpivirine (even better with rilpivirine and new analog non-DAPY dapivirine). Data should be generated for rilpivirine regarding all of the studies.

2) MD simulations are appreciated for analysis of interactions between 18b1 and variants. It would be more compelling if the authors could obtain co-crystal structures of HIV RT variants with 18b1 to confirm the binding interactions identified in the MD simulation.

3) The authors conclude that 18b1 showed a 3-fold potency improvement compared to etravirine. Is this really significant?

4) While many drug-drug interactions are mediated by CYP inhibition, only CYP inhibition was evaluated in this study (and only in vitro studies), there are other mechanisms for drug-drug interactions that do not include CYP inhibition. The authors should clarify that the in vitro data suggests that CYP-mediated drug-drug interactions are less likely for 18b1.

General Comments:

While the compounds are novel, I am unsure that these compounds will be of interest to the rest of the community. There are a number of NNRTIs in design/development, and several FDA approved drugs, but current HIV regimens based on medical guidelines focus on combination therapies that include NRTIs and INSTIs.

The work reported is convincing; additional references (i.e. rilpivirine) and experiments can strengthen the conclusion.

Reviewer #3 (Remarks to the Author):

The manuscript titled "Enhancing Backbone-Binding Interactions: Discovery of Diarylpyrimidine Derivatives bearing Piperazine Sulfonyl as Potent HIV-1 Non-nucleoside Reverse Transcriptase Inhibitors against Wild-Type and F227L/V106A Mutant Viruses" presents some new compounds based on the reported candidates and showed some interesting results. However, the novelty and practical significance is low. I cannot give a positive comment.

1. PDB: 3MEC was used to dock. Why not use the Piperazine Sulfonyl compound co-crystallized with RT?
2. The compounds are still targeting RT? That needs to be further confirmed.
3. The enzyme levels are consistent with the cell based levels or not? Why use the cell-based level result to explain the binding/docking results? It does not make sense.

Reviewers' comments:

Reviewer #1 (Remarks to the Author):

The manuscript entitled “Enhancing Backbone-Binding Interactions: Discovery of Diarylpyrimidine Derivatives bearing Piperazine Sulfonyl as Potent HIV-1 Non-nucleoside Reverse Transcriptase Inhibitors against Wild-Type and F227L/V106A Mutant Viruses” discusses the optimisation of a previous hit compound, BH-11c, ultimately leading to 18b1 through a rather extensive synthetic campaign that was heavily guided by molecular modelling. The final compound, 18b1, demonstrated superior properties to the control compound (ETR), both in terms of potency (especially against problematic resistant strains) and PK properties, which in some categories were superior to that of ETR.

In fact, the only major shortcoming of the compound was found to be its poor oral bioavailability (1.34%). Now specifically because of this poor oral bioavailability I feel the authors should change the final line in their abstract which states “Therefore, we consider compound 18b1 a potential drug candidate for the treatment of HIV-1 infections”, as with such poor oral bioavailability this is not the case unless the route of administration is planned to be IV (which would be suboptimal).

Answer: Thank you for your advice. The relevant description has been revised.

Other than this, the paper is quite well written although it contains many grammatical and technical errors, most of which have been annotated on the attached PDF.

Answer: Thank you for your careful review. The grammatical and technical errors has been corrected.

In my personal opinion, the modelling section of the paper is too lengthy and in fact detracts from the punchy conclusion of the design and synthesis of 18b1, and the authors may consider shortening this section.

Answer: Thank you for your advice. The section of MD simulation has been condensed.

In the supplementary section the procedures are adequately described and the compounds adequately characterised. Therefore I recommend that the article is published after the minor corrections described above and in the attached annotated PDF file.

Reviewer #2 (Remarks to the Author):

Reviewer Summary and Major Claims of the Paper

The authors have designed and synthesized piperazine sulfonyl DAPY analogs as non-nucleoside reverse transcriptase inhibitors (NNRTI). The inhibitors were then evaluated for anti-HIV activity, SAR, inhibition, physicochemical properties such as solubility, milogP, ligand efficiency, and in vitro CYP inhibition.

Compound 18b1 emerged as a top compound showing activity for several variants of HIV RT. 18b1 also showed improved physicochemical properties (compared to NNRTI etravirine) and overall less inhibition by common CYP isoforms. 18b1 was further analyzed through molecular dynamics (MD) studies to analyze binding conformation and drug-target interactions.

Major comments:

1) The authors use etravirine as a standard NNRTI to compare anti-HIV activity, inhibition, and physicochemical properties with the new piperazine sulfonyl DAPY analogs. Why did the authors not include a comparison with rilpivirine? Rilpivirine is another DAPY included in many 2 combination therapies Cabenuva and Complera. It would be more compelling for the authors to compare values for the piperazine sulfonyl DAPY analogs with rilpivirine (even better with rilpivirine and new analog non-DAPY dapivirine). Data should be generated for rilpivirine regarding all of the studies.

Answer: Thank you for your advice. The data of reference drugs rilpivirine and doravirine have been supplemented for comparison.

2) MD simulations are appreciated for analysis of interactions between 18b1 and variants. It would be more compelling if the authors could obtain co-crystal structures of HIV RT variants with 18b1 to confirm the binding interactions identified in the MD simulation.

Answer: Thank you for your advice. The co-crystal structure of 18b1-HIV-1 RT (PDB ID: 8FE8) has been added to gain a thorough understanding of the binding interactions between 18b1 and HIV-1 RT.

3) The authors conclude that 18b1 showed a 3-fold potency improvement compared to

etravirine. Is this really significant?

Answer: Thank you for your advice. The cell-based result is the basis of antiviral potency of NNRTIs. Interestingly, through the medical chemistry strategy of enhancing backbone-binding interactions, a series of novel NNRTIs with improved drug-resistance profiles were identified in this work. In addition, the CYP inhibitory activity and physicochemical properties of these compounds, have also been significantly improved compared with the marketed drugs ETR and RPV. Therefore, we believe that this work is of great guiding significance for the development of a new generation of NNRTIs

4) While many drug-drug interactions are mediated by CYP inhibition, only CYP inhibition was evaluated in this study (and only in vitro studies), there are other mechanisms for drug-drug interactions that do not include CYP inhibition. The authors should clarify that the in vitro data suggests that CYP-mediated drug-drug interactions are less likely for 18b1.

Answer: Thank you for your advice. The relevant description has been revised.

General Comments:

While the compounds are novel, I am unsure that these compounds will be of interest to the rest of the community. There are a number of NNRTIs in design/development, and several FDA approved drugs, but current HIV regimens based on medical guidelines focus on combination therapies that include NRTIs and INSTIs.

Answer: Thank you for your advice. NRTIs are analogues of the natural substrate deoxynucleotide triphosphates, and they inhibit RT as chain terminators, while NNRTIs do not compete for the natural substrate, they bind in NNIBP about 10 Å away from the polymerase active site. Therefore, NNRTIs only act on HIV-1 RT and does not affect human DNA polymerase, with the advantages of high selectivity, specificity and antiviral activity. (*J. Med. Chem.* **2021**, 64, 13604–13621) NNRTIs are key components of HAART, which consists of the most successful antiviral combination therapies Atripla (EFV/emtricitabine/tenofovir disoproxil fumarate) and Complera (RPV/emtricitabine/tenofovir disoproxil fumarate) (*Clinical microbiology reviews*, **2016**, 29, 695-747). Currently, EFV is still the preferred agent of combination therapy

in WHO guidelines, while European guidelines are more inclined to use RPV. In addition, NVP is recommended by WHO as an alternative to first-line drugs, ETR is used primarily in patients with treatment experience who develop resistance to NNRTIs. NNRTIs are widely concerned because of their extensive clinical application, and more than 50 structurally diverse classes NNRTIs have been identified (*J Med Chem.* **2021**; 64:13658-13675). We envisioned that continued research in this area will provide new options for combination therapies.

The work reported is convincing; additional references (i.e. rilpivirine) and experiments can strengthen the conclusion.

Reviewer #3 (Remarks to the Author):

The manuscript titled "Enhancing Backbone-Binding Interactions: Discovery of Diarylpyrimidine Derivatives bearing Piperazine Sulfonyl as Potent HIV-1 Non-nucleoside Reverse Transcriptase Inhibitors against Wild-Type and F227L/V106A Mutant Viruses" presents some new compounds based on the reported candidates and showed some interesting results. However, the novelty and practical significance is low. I cannot give a positive comment.

1. PDB: 3MEC was used to dock. Why not use the Piperazine Sulfonyl compound co-crystallized with RT?

Answer: Thank you for your advice. Unfortunately, there is no co-crystal structure of RT-NNRTIs bearing piperazine sulfonyl at present. Therefore, we confirmed the co-crystal structure of 18b1-RT for the first time in this article, which will provide favorable guidance for further structural optimization.

2. The compounds are still targeting RT? That needs to be further confirmed.

Answer: Thank you for your advice. The co-crystal structure of 18b1-HIV-1 RT (PDB ID: 8FE8) has been added in this manuscript, indicating 18b1 bonded to the NNRTI binding pocket (NNIBP). Besides, the difference of antiviral activity against wild-type and mutant strains can also indirectly indicate that the compounds act on the NNIBP. Amino acid mutations in NNIBP could affect the antiviral activity of NNRTIs (reference drugs efavirenz, nevirapine and etravirine), but not NRTIs (reference drug zidovudine). It is obvious that the antiviral activities of these compounds against

various variants are very different.

3. The enzyme levels are consistent with the cell based levels or not? Why use the cell-based level result to explain the binding/docking results? It does not make sense.

Answer: Sorry to confuse you. In fact, the cell-based antiviral activities of NNRTIs were inconsistent with their enzyme-inhibitory potencies to some extent. These differences are considered to be due to the variations in the HIV-1 RT-substrate binding affinities and polymerase processivity on different nucleic acid templates, and have been observed in most NNRTI series (*J. Biol. Chem.* **1993**, 268, 276-281; *J. Med. Chem.* **2020**, 63, 4837-4848; *J. Med. Chem.* **2016**, 59, 7991-8007). Besides, the EC₅₀ and IC₅₀ values are very much depending on the conditions of testing. For the MTT method in cell activity testing, the EC₅₀ values depends on the number of cells, the quantity of virus (and the ratio of the number of virus particles/number of cells), the replication rounds, and the concentration of (serum) proteins in the medium (indicative for protein binding of the compound). However, for the RT assay, the IC₅₀ values depend on the amounts of the template/primer and enzyme. Also, the differences between the two assay systems can be attributed at least in part to differences in the physiochemical properties (especially the intrinsic solubility and membrane permeability).

For the second question, compared with enzyme-inhibition activity, cell-based antiviral screening can better reflect the real activity in the human body, so we focus on antiviral activity at the cellular level. The amino acid mutations in NNIBP can significantly affect cell-based antiviral activity of NNRTIs, the most likely reason is amino acid mutations could change the interactions between ligands and protein. Therefore, the determination of co-crystal structure, molecular docking and molecular dynamics simulation studies were conducted to understand the binding modes of compounds and NNIBP, which is used to explain why the compounds have different activities against HIV-1 RT various variants, instead of using the cell-based level result to explain the binding/docking results.

REVIEWERS' COMMENTS:

Reviewer #3 (Remarks to the Author):

The revision looks nice to me, I would love to suggest "Publish as is".

REVIEWERS' COMMENTS:

Reviewer #3 (Remarks to the Author):

The revision looks nice to me, I would love to suggest "Publish as is".

Answer: Thank you for your advice.